# A Channel Coding Benchmark for Meta-Learning

**Rui Li**
Samsung AI Center
Cambridge, UK
`rui.li@samsung.com`

**Ondrej Bohdal**
Samsung AI Center and
University of Edinburgh, UK
`ondrej.bohdal@ed.ac.uk`

**Rajesh Mishra**
UT Austin, USA
`rajeshkmishra@`
`austin.utexas.edu`

**Hyeji Kim**
UT Austin, USA
`hyeji.kim@`
`austin.utexas.edu`

**Da Li**
Samsung AI Center
Cambridge, UK
`da.li1@samsung.com`

**Nicholas Lane**
Samsung AI Center
Cambridge, UK
`nic.lane@samsung.com`

**Timothy Hospedales**
Samsung AI Center and
University of Edinburgh, UK
`t.hospedales@ed.ac.uk`

## Abstract

Meta-learning provides a popular and effective family of methods for data-efficient learning of new tasks. However, several important issues in meta-learning have proven hard to study thus far. For example, performance degrades in real-world settings where meta-learners must learn from a wide and potentially multi-modal distribution of training tasks; and when distribution shift exists between meta-train and meta-test task distributions. These issues are typically hard to study since the shape of task distributions, and shift between them are not straightforward to measure or control in standard benchmarks. We propose the *channel coding* problem as a benchmark for meta-learning. Channel coding is an important practical application where task distributions naturally arise, and fast adaptation to new tasks is practically valuable. We use our MetaCC benchmark to study several aspects of meta-learning, including the impact of task distribution breadth and shift, which can be controlled in the coding problem. Going forward, MetaCC provides a tool for the community to study the capabilities and limitations of meta-learning, and to drive research on practically robust and effective meta-learners.

## 1 Introduction

Meta-learning, or learning-to-learn, aims to provide data-efficient learning of new tasks by training improved learning algorithms using a distribution over tasks. The promise of such data efficient learning has long inspired research [32, 35], and recently grown into a thriving research area in which rapid progress is being made [8, 41, 9, 13]. While performance has improved steadily, particularly on standard image recognition benchmarks, several fundamental outstanding challenges have been identified [13]. Notably, state of the art meta-learners have been shown to suffer in realistic settings [40, 37] when required to generalize across a diverse rather than artificially narrow range of tasks – i.e. the task distribution is broad and multi-modal; and when there is distribution shift between the (meta)training and (meta)testing tasks. These conditions are almost inevitable in real-world applications where, for example, robots should generalize across the range of manipulation tasks of interest to humans [40], and image recognition systems should cover a realistically wide range

of image types [37]. However, systematic study of these issues is hampered because conventional benchmarks do not provide a way to quantitatively measure or control the complexity or similarity of task distributions: Does an image recognition benchmark covering birds and airplanes provide a more or less complex task distribution to meta-learn than one covering flowers and vehicles? Is there greater task-shift if a robot trained to pick up objects must adapt to opening a drawer or throwing a ball? In this paper, we contribute to the future study of these issues by introducing a *channel coding* meta-learning benchmark termed MetaCC, which enables finer control and measurement of task-distribution complexity and shift.

Channel coding is a classic problem in communications theory of how to encode/decode data to be transmitted over a capacity limited noisy channel so as to maximize the fidelity of the received transmission. While there is extensive theory on optimal codes for analytically tractable (e.g., Gaussian) channels, recent work has shown that codecs obtained by deep learning provide clearly superior performance on more complex challenging channels [17, 16]. In this paper, we focus on learning the *decoder* for a fixed encoder[1]. Best deep channel coding however is achieved by training codecs tuned to the noise properties of a given channel. Thus, a highly practical meta-learning problem arises: Meta-learning a channel code learner on a distribution of training channels, which can rapidly adapt to the characteristics of a newly encountered channel. By way of example, the role of meta-learning is now to enable the codec of a user's wireless mobile device to rapidly adapt for best reception as she traverses different environments or switches on/off other sources of interference.

We introduce channel coding problems [16, 17] as tasks to study the performance of meta-learners, defining the MetaCC benchmark to complement existing ones [40, 37]. Our benchmark spans five channel families, including a real-world measurement of channel based on software defined radio (SDR). We show how the channel coding problem uniquely leads to natural model-agnostic ways to measure the *breadth* of a task distribution, as well as the *shift* between two task distributions (Fig. 1) – quantities that are not straightforward to measure in vision benchmarks. Building on these metrics, we use MetaCC to answer the following questions, among others:

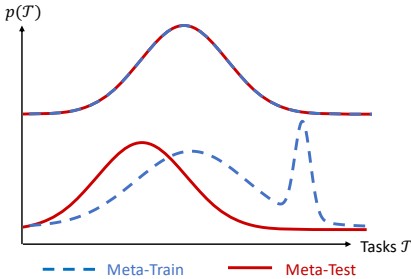

**Figure 1:** Schematic illustration of meta-learning scenarios. Top: The typical assumption of $p_{tr}(\mathcal{T}) = p_{te}(\mathcal{T})$ is rarely met in practice. Bottom: (i) Given a complex distribution of training tasks, meta-learners may under-fit by failing to provide fast adaptation to all modes in the distribution. (ii): Realistic scenarios pose distribution shift between training $p_{tr}(\mathcal{T})$ and deployment $p_{te}(\mathcal{T})$.

**Q1:** *How vulnerable are existing meta-learners to underfitting when trained on complex task distributions?* Existing studies [40, 39] have identified this as a challenge but have not been able to study it systematically without task complexity measures. **Q2:** *How robust are existing meta-learners to task-distribution shift between meta-train and meta-test task distributions?* This challenge has been widely observed in both robotics [40] and computer vision [37, 12] but has not been able to be measured without task-distribution distance measures. **Q3:** *How much can meta-learning benefit in terms of transmission error-rate on a real radio channel?* Deep learning powered codecs specifically trained with canonical channels have shown improved performance over traditional codecs [17, 29], and there are applications of meta-learning to simpler tasks than channel decoding in comms e.g. demodulation [28, 5]. However, it is yet to be determined how well can meta-learners perform in a transition from simulation to real world communication channels.

## 2 Background

### 2.1 Channel Coding Background

Channel coding is a key element in a communication system. Its role is to introduce controlled redundancy so that the receiver can reliably and efficiently recover the message from a *corrupted* received signal. A typical channel coding system consists of an encoder and a decoder, as illustrated in Fig. 2. In this example a rate 1/2 channel encoder maps $K$ message bits $\mathbf{b} \in \{0,1\}^K$ to a length-$2K$ transmitted signal $\mathbf{c} \in \{-1,1\}^{2K}$. In a more general setting a rate $1/r$ encoder maps

---

[1]This is the practically relevant setting as communication standards defining the encoding protocol are not easy to change, but decoders can be upgraded without changing the standard.

$\mathbf{b} \in \{0,1\}^K$ to $\mathbf{c} \in \{\pm1\}^{rK}$. The signal $\mathbf{c}$ is then transmitted with the noise effect experienced by the signal in the communication medium described by conditional distribution $p(\mathbf{y}|\mathbf{c})$, and channel outputs a noisy signal $\mathbf{y} \sim p(\mathbf{y}|\mathbf{c}), \mathbf{y} \in \mathbb{R}^{2K}$. A canonical example is Additive White Gaussian Noise (AWGN) channels, where $\mathbf{y} = \mathbf{c} + \mathbf{z}$ for Gaussian $\mathbf{z} \sim \mathcal{N}(0, \sigma^2 I)$. The decoder in turn takes the noisy signal as input and estimates the original message, i.e. $\hat{\mathbf{b}} = f_\theta(\mathbf{y}) \in \{0,1\}^K$. The reliability of an encoder/decoder pair is measured by the probability of error, such as Bit Error Rate (BER) defined as $\sum_{k=1}^K \mathbb{P}(\hat{b}_k \neq b_k)$. We treat the decoding problem as a $K$-dimensional binary classification task for each of the ground-truth *message bits* $b_k$.

**Neural Decoder for Convolutional Codes**  We focus on learning a decoder for a fixed rate 1/2 convolutional encoder which maps $\mathbf{b} \in \{0,1\}^K$ to $\mathbf{c} \in \{-1,1\}^{2K}$ according to $c_{2k} = 2(b_k + b_{k-1} + b_{k-2}) - 1$, $c_{2k+1} = 2(b_k + b_{k-2}) - 1$ for $k \in [1:K]$ assuming $b_0 = b_{-1} = 0$ (also illustrated in Appendix A). The sequential nature of convolutional encoding naturally aligns with convolutional neural networks. Practically, reliable and efficient decoders form an essential part of almost all kinds of communication systems, from wireline to wireless communications including both Wi-Fi and cellular. Thus there has been significant interest in applying deep learning to improve channel decoding (and coding itself) [26].

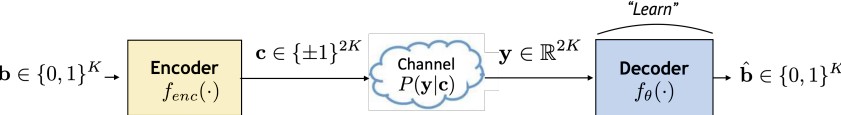

**Figure 2:** An illustration of the channel coding problem. We learn a channel decoder for a fixed encoder under various channel models.

**Adaptive Neural Decoder**  The channel $p(\mathbf{y}|\mathbf{c})$ can vary over time, and is unknown to the decoder. To help the decoder estimate the channel, pilot signals that are known messages $\mathbf{b}_{known}$ are sent to the decoder before the transmission begins, so that the decoder can extract channel information from $\mathbf{y}$ and $\mathbf{b}_{known}$. When modeling the decoder as a neural network instead of an analytical algorithm, one trains the decoder for a specific channel using pairs $(\mathbf{y}, \mathbf{b}_{known})$ with pilot signals as ground-truths and their corresponding noisy received values as inputs. The optimization goal is to minimize a loss $\mathcal{L}$, which is typically in form of binary cross-entropy, with respect to decoder $f_\theta$ as

$$\theta^* = \mathrm{argmin}_\theta \, \mathbb{E}_{\mathbf{b}_{known}, \mathbf{y}} \mathcal{L}(\mathbf{b}_{known}, f_\theta(\mathbf{y})) \tag{1}$$

To ensure good performance as channel characteristics $p(\mathbf{y}|\mathbf{c})$ change due to e.g. weather or moving users, which always happen in realistic communications, the neural decoder $f_\theta$ should adapt to evolving channel. Meta-learning is therefore a promising tool to enable rapid decoder adaptation with few pilot codes, as confirmed by early evidence [15]. Conversely, channel coding provides a lightweight benchmark for contemporary meta-learners, allowing control of the task complexity and distribution-shift, thanks to the mathematical representability and tractability of channel models.

**Connection to Standard Benchmarks**  To clarify the connection to common vision benchmarks: Unique messages $\mathbf{b}$ correspond to image categories, with noisy signals $\mathbf{y}$ corresponding to individual images to recognize. The channel model $p(\mathbf{y}|\mathbf{c}(\mathbf{b}))$ corresponds to the generative process for images conditional on a category; and our learned decoder $f_\theta(\mathbf{y})$ corresponds to an image recognition model. Uniquely, we can control the generation process for data $p(\mathbf{y}|\mathbf{c}(\mathbf{b}))$ which is not feasible for images.

## 2.2  Meta-Learning

Meta-learning usually considers distributions over tasks for training and testing $p_{tr}(\mathcal{T})$ and $p_{te}(\mathcal{T})$. Each task $\mathcal{T}_i$ is associated with a dataset $D_i = \{\mathbf{x}_i^j, \mathbf{y}_i^j\}_{j=1}^J$, which we split into $D_i = D_i^{tr} \cup D_i^{val}$. We are interested in learning models $f_\theta$ of the form $\hat{\mathbf{y}} = f_\theta(\mathbf{x})$ using some algorithm $\mathcal{A}$ that minimizes a loss function $\mathcal{L}(\theta, D)$ on data $D$ with respect to parameters $\theta$. The algorithm itself is paramaterized by meta-parameter $\phi$, i.e., $\theta^* = \mathcal{A}(D, \mathcal{L}, \phi)$. The goal of meta-learning is to find the parameters $\phi$ of algorithm $\mathcal{A}$ that lead to strong validation performance after learning.

$$\phi^* = \mathrm{argmin} \, \mathbb{E}_{\substack{\mathcal{T} \sim p(\mathcal{T}) \\ (\mathcal{D}^{tr}, \mathcal{D}^{val}) \in \mathcal{T}}} \mathcal{L}(\mathcal{A}(D^{tr}, \mathcal{L}, \phi), D^{val}) \tag{2}$$

When datasets $D^{tr}$ are small, this leads to meta-optimization for a data-efficient learner, as pioneered by MAML [8], which chooses meta-parameter $\phi$ as the initial condition of the optimization for $\theta$ by $\mathcal{A}$. Once meta-learning is complete, we can draw a new task $\mathcal{T}' \sim p_{te}(\mathcal{T})$, and solve it efficiently as

$$\theta^* = \mathcal{A}(D', \mathcal{L}, \phi^*). \tag{3}$$

# 3 MetaCC: A Coding Benchmark for Meta-Learning

**Constructing Task Distributions**  We consider five *families* of channel models and corresponding decoding tasks. These include synthetic Additive White Gaussian Noise (AWGN), Bursty, Memory noise, and Multipath interference channel used by 3GPP and ITU to decide which codecs to use in 4G LTE and 5G communication standards. Furthermore, we consider a final family consisting of data recorded from a real software-defined radio testbed. See Appendix B for details. Each family is analogous to a dataset in common multi-dataset vision benchmarks [37]. All four synthetic channel families are used for meta-training, and the real wireless channel is held out for meta-testing.

To define task distributions, we consider uni-modal and multi-modal settings. In the *single-family, uni-modal* case, a task distribution $p$ corresponds to a specific channel class as discussed above, paramaterized by continuous channel parameters $\omega$ (e.g., the variance of additive noise or multipath strength). The distribution of tasks in this family then depends on the prior over channel parameter $\omega$, $p(\mathcal{T}) = \int_\omega p(\mathcal{T}|\omega)p(\omega)$. We can control the width of a task distribution by varying the width of the, e.g. uniform distributed, prior $p(\omega)$. In the *multi-family, multi-modal* case we can define a more complex task distribution as a mixture over multiple channel types $p_k$, each with its own distribution over channel parameters $\omega$, $p(\mathcal{T}) = \sum_k \int_\omega \pi_k p_k(\mathcal{T}|\omega)p_k(\omega)$.

**Quantifying Task Distribution Shift and Breadth**  We quantify the train-test task shift distance (Definition 1) and diversity of each task (Definition 2), based on information theoretic measures. In a coding benchmark, we can control these scores by choosing appropriate set of channel models, which allows us to evaluate the variability of meta-learning with the task distribution breadth and shift as illustrated in Fig. 1. We demonstrate this in Section 4 (Fig. 4, 5).

**Definition 1 (Train-Test Task-Shift $S(p_a(\mathcal{T}), p_b(\mathcal{T}))$)**  Simply *measuring* the shift between training and testing task distributions has previously been an open problem in meta-learning. However this becomes feasible to define for the channel coding problem. We quantify the distance between a test distribution $p_a(\mathcal{T})$ and a training distribution $p_b(\mathcal{T})$ using the Kullback–Leibler divergence (KLD) a.k.a. the relative entropy [19]. The KLD-based shift distance score is defined as:

$$S(p_a(\mathcal{T}), p_b(\mathcal{T})) := \mathbb{E}_{\mathbf{c}}[D_{KL}(p_a(\mathbf{y}_a|\mathbf{c})||p_b(\mathbf{y}_b|\mathbf{c}))] + \mathbb{E}_{\mathbf{c}}[D_{KL}(p_b(\mathbf{y}_b|\mathbf{c})||p_a(\mathbf{y}_a|\mathbf{c}))], \tag{4}$$

where $p_a(\mathbf{y}_a|\mathbf{c})$ and $p_b(\mathbf{y}_b|\mathbf{c})$ denote the channels associated with $\mathcal{T}_a$ and $\mathcal{T}_b$, respectively. The distance is large if a testing distribution $p_a$ introduces a very different distribution over received messages $\mathbf{y}$ for a given code $\mathbf{c}$ compared to training $p_b$, and zero if they induce the same distribution. One can also consider asymmetric KLD, i.e., $\mathbb{E}_{\mathbf{c}}[D_{KL}(p_a(\mathbf{y}_a|\mathbf{c})||p_b(\mathbf{y}_b|\mathbf{c}))]$ (See Appendix E).

**Definition 2 (Diversity Score $D(\mathcal{T})$)**  The diversity score of a task distribution $p(\mathcal{T})$ is defined as mutual information between the channel parameter $\omega$ and the received signal $\mathbf{y}$:

$$D(\mathcal{T}) = \mathbb{E}_{\mathbf{c}}[I(\omega; \mathbf{y}|\mathbf{c})],$$

where $\omega$ denotes the channel parameter (latent variable) for the task distribution, i.e., $p(\mathbf{y}|\mathbf{c}) = \int_\omega p(\mathbf{y}|\mathbf{c}, \omega)p_\omega(\omega)$. We will see that this metric will quantify amenability to meta-learning. Intuitively, decoding benefits more from meta-learning when the channel distribution $p(\mathbf{y}|\mathbf{c}, \omega)$ differs more across tasks (channel parameter $\omega$). I.e., knowing the task conveys more information about $\mathbf{y}$.

**Estimation of Scores**  In order to estimate shift-distance and diversity scores, we generate samples according to the corresponding task distributions and estimate each of these scores via Kraskov-Stögbauer-Grassberger (KSG) estimator [18, 34].

**Discussion**  While we denote each channel to learn as a '*task*', we note that in our application each task shares the same label-space of messages to recognize. As such our goal could also be understood as few-shot supervised adaptation to new *domains* by meta-learning. Thus our evaluation will compare to the simple domain generalization baseline of conventionally learning a decoder on all data from $p_{tr}(\cdot)$ and applying it directly to tasks in $p_{te}(\cdot)$ without adaptation.

# 4 Experiments

We first evaluate the impact of training distribution diversity on meta-learning performance, followed by that of train-test task distribution shift.

**Dataset and Task Design**   We consider a wide range of channel scenarios that are described in Appendix D. To facilitate evaluation, we create a dataset of (received noisy) codewords and multiple transmitted messages under each channel model. For each of the benchmark scenarios, we created a dataset with 100 randomly sampled noise setups $\omega$, from the noise family specific to the scenario. Each noise setup has 1000 randomly generated true codes ('classes') with 20 examples (noisy received messages) for each type. When generating a meta-training task, we randomly sample a noise set-up and then sample $N = 5$ codes with $K = 5$ support examples and $L = 15$ target or query examples. Note that the support and target messages are independently sampled, hence unlikely to overlap. This makes the tasks similar to the standard $N$-way $K$-shot problems, but instead of a class-adaptation problem, we solve a fast domain-adaptation problem. For meta-testing, we have another dataset with 50 manually specified noise setups. For each noise setup we randomly generate 100 messages and with 50 examples each. The meta-testing dataset is shared across various scenarios. Meta-testing tasks are generated in the same way as meta-training tasks. All of the datasets are small enough to easily fit into the GPU memory, allowing fast experimentation.

**Meta-Learning Algorithms**   We have evaluated a variety of meta-learning approaches[2]. These include gradient-based learners **MAML** [8], its first order approximation **MAML FO**, **Reptile** [24], **ANIL** [30], **MetaSGD** [23], **KFO** [3], **MetaCurvature** [27] **CAVIA** [41] **BOIL** [25]; and feed-forward learners **ProtoNets** [33], and **MetaBaseline** [4]. See Appendix F for details. These adaptive methods are compared to the standard non-adaptive approach of **empirical minimization (ERM)**, which trains a conventional neural decoder on the union of meta-training tasks, and has been shown to be a strong baseline [21, 11]. We further include a non-meta-few-shot baseline SUR [7] in both its original form that builds on ProtoNet, i.e. **SUR PROTO**, and a novel version **SUR ERM** that instead builds on ERM. All these neural approaches are compared to the classic non-neural **Viterbi** decoder [10]. This maximum likelihood based algorithm, which is known to achieve close-to-optimal block error rate under the simplest AWGN channel. We have extended the implementations provided by *learn2learn* library [2] under the MIT License. Note that channel decoding is a *multi-label* problem that requires predicting a *vector* of bits for each input example, rather than a single multi-class classification. While this is straightforward for gradient-based meta-learners, we extended the implementation of the feed-forward meta-learners to support this.

**Hyperparameters and Architecture**   All neural approaches used the same hyperparameters and CNN architecture for consistency. We used Adam optimizer with a meta-learning rate of 0.001 for the outer-loop, SGD with fine-tuning learning rate of 0.1 for the inner-loop consisting of 2 adaptation steps, 10 tasks in a meta-batch and 80000 meta-training iterations. Each task consisted of 5 different adaptation types ('classes'), with 5 support and 15 target examples. The ground truth messages are 10 bits long, encoded by the $1/2$ rate convolutional encoder. Hence the message input to the decoder has shape $1 \times 10 \times 2$. We fix the decoder architecture as a CNN with 4 layers, 64 filters, kernel size 3, and stride $(1, 2)$. The CNN is followed by a linear fully-connected layer of size $64 \times 1$.

## 4.1 Impact of Training Distribution Diversity on Meta-Learning Performance

In this first section we investigate how meta-learners cope with task distributions of varying breadth and complexity, since previous studies have suggested that capacity could be a limiting factor of existing meta-learners [40, 31, 39]. We would like meta-learners to be capable of learning from a broad range of auxiliary tasks, without requiring the auxiliary task distribution to be carefully constructed in advance for similarity to each given target task (cf, Fig. 1).

**Setup**   In these experiments, we fix the meta-testing task distribution $p_{te}(\mathcal{T})$ to ensure comparability, and then evaluate performance when the training distribution $p_{tr}(\mathcal{T})$ is focused around the testing condition '*focused*' vs when it is spread more broadly around the testing distribution '*expanded*'. To expand the training distribution in the single-family/uni-modal case, we use a wider prior on the channel parameter $p_{tr}^{expand}(\omega) = \text{Unif}(a - \delta, b + \delta)$ vs $p_{te}(\omega) = \text{Unif}(a, b)$ when constructing task distributions as discussed in Section 3. In the multi-modal case, we use a single channel family for

---

[2]Our code repository is publicly available at https://github.com/ruihuili/MetaCC.

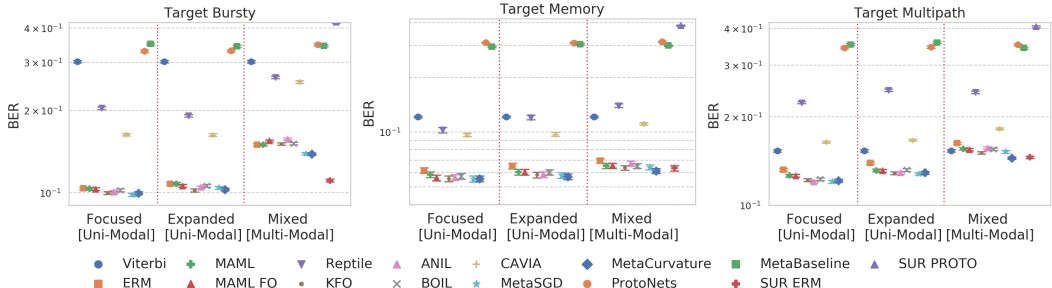

**Figure 3:** Meta-testing on Bursty, Memory and Multipath task families (subplots). Y-axis is Bit Error Rate (BER, lower is better). Bars indicate meta-test standard-errors. 'Focused' and 'Expanded': Within-family – uni-modal training distributions focused or widely distributed around the testing distribution. 'Mixed' column: Across-family setting where the training distribution is a mixture of all four task families including the testing distribution. Performance is impacted by increased diversity in the training distribution (compare focused→mixed).

$p_{te}$ and a multi-modal mixture ('*mixed*') of families for $p_{tr}$ composed of all the synthetic channel models (Appendix B) including the testing distribution $p_{te}$.

**Results**    Fig. 3 summarizes the results for meta-learning on training distributions of varying widths for both uni-modal (marked as 'focused' and 'expanded') and multi-modal (marked as 'mixed') conditions, and for different target channels (four subplots). N.B. we plot for each learner the aggregated performance over a range of noise settings for each target family type, while the disaggregated version can be found in Fig. 10 in Appendix. From the results, we can see that: (i) Most neural models outperform the industry standard neural decoder on realistic complex channel models (bursty, memory, multipath). (ii) The best meta-learners surpass the non-adaptive ERM baseline especially on the challenging multi-path dataset. (iii) In the within-family case (left groups), focusing the meta-training distribution on the meta-testing condition (x-axis: focused) vs a diverse meta-training regime (x-axis: expanded) does not visibly affect meta-testing performance on our log-performance scale. In the across-family case (right, x-axis: mixed), transferring from a multi-modal training distribution to a specific testing distribution incurs a visible difference to performance for bursty and multi-path target channels. This confirms that meta-learner capacity for fitting a multi-modal training distribution *does* impact performance [39, 31]. (iv) Where applicable (mixed), the original SUR PROTO is out-performed by peer learners, while our modified version SUR ERM exceeds the rest.

To further understand these results we compute the breadth of each training regime as measured by its *diversity* (Section 3). Note that we can measure the diversity of both focused/expanded (uni-modal) and mixed (multi-modal) training regimes with the same metric. As expected, the mixed regimes lead to higher diversity. Fig. 4 plots the margin between meta-learners and the vanilla non-adaptive baseline against the diversity score of the channel. We can see that, while more diverse training regimes reduce absolute performance (Fig. 3), the benefit provided by meta-learning over vanilla ERM can increase with more diverse training regimes (Fig. 4).

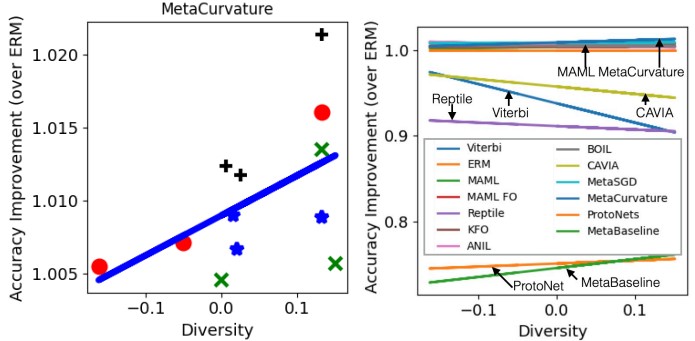

**Figure 4:** Correlation between task distribution diversity score and benefit of meta-learning. **Left**: Example fitting for MetaCurvature. X-axis: Diversity score of the channel; Y-axis: Accuracy gain over ERM. Symbols indicate different target channels from Fig. 3 (o: AWGN, x: Bursty, *: Memory, +: Multipath). **Right**: Fitted lines for all meta-learners. Meta-learning can provide greater benefit on more diverse task distributions. Plots for all meta-learners are available in Appendix G.

Intuitively, the more the channel configuration parameters determine

the output message distribution, the more potential benefit there is from meta-learning how to adapt to a given channel.

## 4.2 Impact of Train-Test Distribution Shift on Meta-Learning Performance

**Within-Family Setup** We first illustrate the uni-modal within task family case, where we create distribution shift by setting $p_{tr}(\omega) \neq p_{te}(\omega)$. Specifically, we define two training task distributions using two different non-overlapping uniform priors on $\omega$ corresponding to different channel SNR-Bs (denoted 'high' and 'low') in the Bursty channel. We then train meta-learners on each, and evaluate them on a range of task parameters $\omega$ that are both in- and out-of-domain with respect to the training distribution. In this section, we study how each of the baseline learners' performance depends on distribution shift between training $p_{tr}(\mathcal{T})$ and testing $p_{te}(\mathcal{T})$ task distributions.

**Results** Fig. 5 (left) shows the results of generalizing across a range of meta-testing tasks (x-axis), for models learned within each of the two specified training domains. Note that the 'difficulty' of the shown testing tasks is non-uniform i.e. higher SNR-B tasks are easier. This means that other things being equal we expect worse performance toward the left of the graphs; and that the models trained on the 'Low' SNR range (blue) and models trained on the 'High' SNR-B range (orange) have been exposed to the hardest/easiest training regime respectively. Concretely, the meta-learner's performance is clearly better when operating within-domain than when operating with train-test distribution shift, indicated by the crossing of the lines corresponding to the two training conditions.

**Across-Family Setup** We next consider the more challenging across-family setting. In this case we create task distributions defined by each channel type and use them for meta-training. We then consider several channel types for meta-testing and evaluate pairs of matched and mis-matched train/test regimes. The four synthetic families are used for meta-training, and all five including our *real-world* channel dataset (See Appendix C for implementation details) are used for meta-testing. This setup aligns with the sim-to-real paradigm that is widely applied in other machine learning applications such as robotics and vision [36, 14] since it is easier to conduct large scale training on simulated data, and evaluate efficient-adaptation on sparser real-world data. We are the first to consider and benchmark meta-adaptation as a solution for sim-to-real transfer in channel coding.

**Results** From the results in Fig. 6, we can see that: (i) All learned models generally perform best in the within-family conditions; IE: when source channel on the x-axis matches the target channel of the sub-plot, as indicated by the dashed box. (ii) Some across-channel family conditions also perform quite well, such as Multipath $\rightarrow$ AWGN; but not others, such as Bursty $\rightarrow$ AWGN. However, some specific channel families such as Bursty cannot be successfully addressed when transferring from any other cross-family training distribution. Overall the results show that robustness to distribution-shift is an issue for both non-adaptive and meta-learned adaptive decoders.

**Meta-Learned Decoders on a Real Wireless Channel** Notably, the results in Fig. 6 (bottom right) confirm confirm that while the neural but non-adaptive ERM decoder fails to reliably outperform Viterbi, the best adaptive neural codes clearly outperform Viterbi. In particular the best meta-learners provide a 58% and 30% reduction in error rate compared to the standard neural decoder (ERM) and classic Viterbi decoder respectively. This shows the potential of meta-learning for improving the performance of future real-world comms systems. It also shows benchmark's value, as advancements in meta-learning driven by the benchmark can translate directly to real-world impact.

**Summary** We have seen in the previous two experiments that meta-learning performance is best when $p_{te} = p_{tr}$, with performance degrading smoothly when there is small deviation between them (Fig. 5 (left)), and sometimes dropping significantly when they are entirely different task families (Fig. 6). A key feature of channel coding as a meta-learning benchmark is the ability to measure the distance between task distributions in a systematic manner, as explained in Section 3. We can thus aggregate our results across experiments and plot normalized accuracy against meta-train meta-test task distribution distance as illustrated for MetaCurvature in Fig. 5 (middle) (and in Appendix H). Here, each dot on the scatter plot is an experiment. In Fig. 5 (right), we compare the fitted performance curves for each meta-learner. We can see that they all have a positive relative accuracy slope: providing more benefit over vanilla ERM by adapting to increasing train-test distribution shift. Going forward, the evaluation shown in Fig. 5 (mid., right) can provide a metric to benchmark the performance of meta-learners under train-test distribution shift.

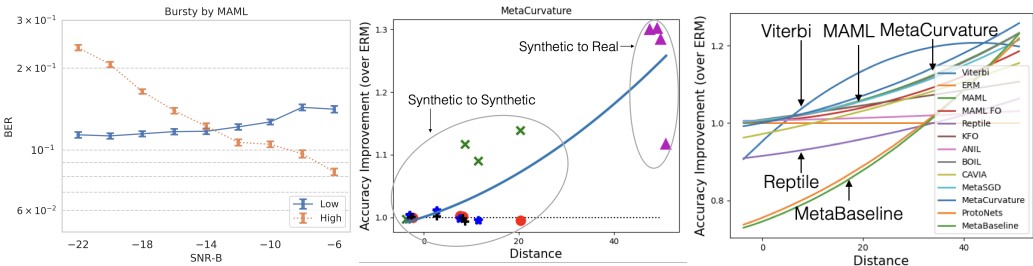

**Figure 5: Left:** Impact of train-test distribution shift on decoding performance, within-family condition. X-axis: Meta-*testing* distribution parameter. Curves: Meta-training regime. 'Low' training regime corresponds to lower SNR sampled from range (-2.5,3.5) and SNR-B from (-23, -17) for Bursty channel, and 'High' training regime corresponds to SNR and SNR-B sampled from (8.5,13.5) and (-11, -5), respectively. Performance of meta-learners degrades relatively smoothly as decoders are evaluated in increasingly out-of-domain conditions (crossing high/low lines). **Mid.:** Impact of train-test task distribution distance on decoding accuracy. X-axis: The KL distance score between train and test distributions (Eq. 4). Meta-learners provide greater improvement with distance. Y-axis: accuracy gain over ERM. Adaptive decoding with MetaCurvature, where red (o), green (x), blue (∗), black (+), and purple (△) color corresponds to AWGN, Bursty, Memory, Multipath, and Real target channels, respectively. **Right:** Fitted curves for performance gain over ERM as a function of distance score. Scatter plots for all meta learners and the fitted curves are available in Appendix H.

## 4.3 Comparison of Meta-Learners

**Table 1:** Aggregate comparison of all meta-learners across all experiments. **Top**: Percentage of runs where each algorithm significantly outperform ERM (p-values < 0.05, higher is better). **Mid.**: Percentage of times when an algorithm significantly outperform Viterbi. **Bottom**: Average rank of each meta-learner across all runs (Lower is better). **R**: Sim-to-real and **S**: Sim-to-sim.

| | R/S | Viterbi | ERM | MAML | FOML | Reptile | KFO | ANIL |
|---|---|---|---|---|---|---|---|---|
| vs ERM ↑ | R | 87.6 | N/A | 94.1 | 95.4 | 46.4 | 68.0 | 37.9 |
| vs ERM ↑ | S | 47.7 | N/A | 26.1 | 31.4 | 2.6 | 36.9 | 27.1 |
| vs Viterbi ↑ | R | N/A | 4.2 | 46.4 | 34.3 | 0.0 | 10.5 | 4.6 |
| vs Viterbi ↑ | S | N/A | 38.2 | 42.8 | 42.5 | 8.5 | 43.8 | 43.5 |
| Rank ↓ | R | 5.3±0.2 | 11.6±0.1 | 4.1±0.1 | 5.6±0.1 | 11.4±0.1 | 10.1±0.1 | 11.6±0.1 |
| Rank ↓ | S | 5.4±0.2 | 7.7±0.1 | 5.8±0.1 | 5.2±0.1 | 10.7±0.0 | 4.4±0.2 | 5.3±0.1 |

| | R/S | BOIL | CAVIA | MetaSGD | MetaCu. | ProtoNe. | MetaBa. |
|---|---|---|---|---|---|---|---|
| vs ERM ↑ | R | 95.4 | 73.5 | 95.1 | 96.1 | 83.7 | 87.9 |
| vs ERM ↑ | S | 30.1 | 6.2 | 49.0 | 47.4 | 1.3 | 1.0 |
| vs Viterbi ↑ | R | 47.4 | 19.0 | 41.5 | 55.6 | 28.4 | 31.0 |
| vs Viterbi ↑ | S | 42.8 | 12.1 | 45.1 | 47.1 | 1.3 | 0.0 |
| Rank ↓ | R | 3.7±0.1 | 8.1±0.1 | 5.2±0.1 | 2.7±0.1 | 6.2±0.1 | 5.3±0.2 |
| Rank ↓ | S | 4.8±0.1 | 9.6±0.1 | 3.8±0.1 | 3.4±0.1 | 12.3±0.0 | 12.6±0.0 |

Given the experiments so far, we can answer the question of which meta-learners are best (and worst) for adaptive channel coding. Aggregating across all the previous experiments, we evaluate two metrics: (i) The *percentage of wins* vs the natural baseline of non-adaptive neural ERM. Where a win is computed by a statistically significant ($p < 0.05$) improvement of each competitor vs ERM with respect to one experiment (train and test channel condition). (ii) The *average rank* of each competitor when ranking their accuracy in each experiment.

The results in Tab 1 show that: (i) MetaSGD and MetaCurvature provide the best performance overall. (ii) ANIL and KFO are the least competitive meta-learners in sim-to-real, and ProtoNets

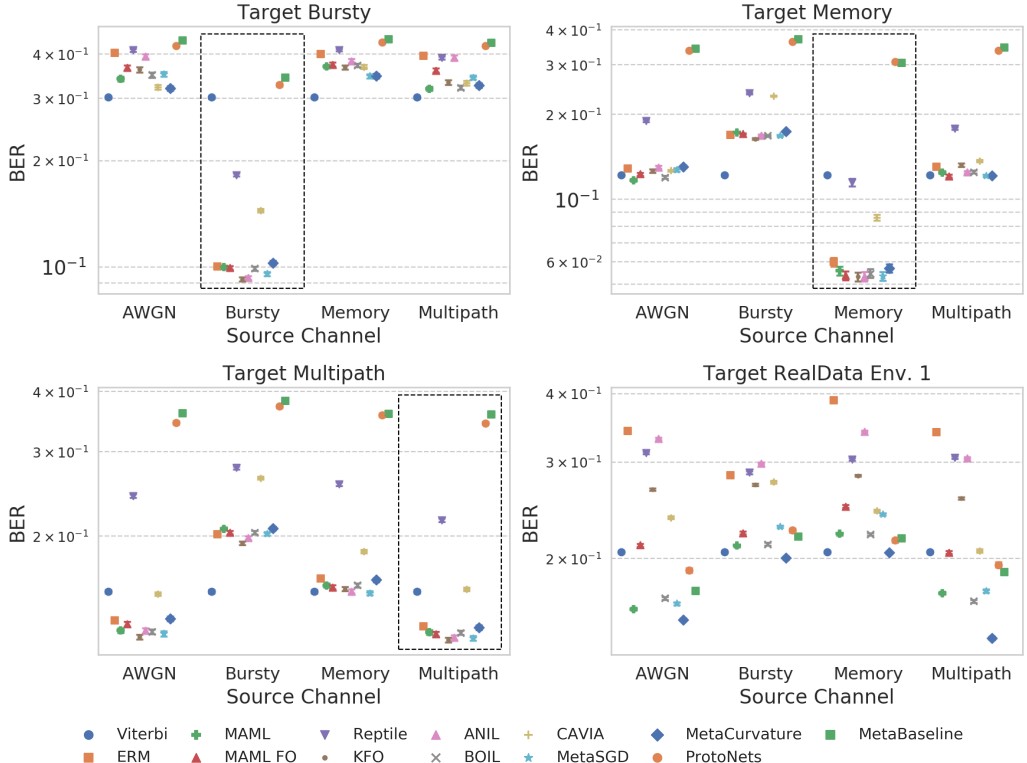

**Figure 6:** Impact of train-test distribution shift on decoding performance. Each sub-plot shows results of meta-testing on one of the AWGN, Bursty, Memory, Multipath task families, after learning on different meta-training channels (line groups, x-axis). Lines correspond to meta-test standard errors of each algorithm. Boxes indicate when meta-training and meta-testing task families align, i.e., the within-family condition. Overall meta-learning works well within task distribution (boxes), and sometimes across task distribution.

and MetaBaseline under-perform the most in sim-to-sim. Reptile is among the weakest overall. (iii) The feed-forward learners are not particularly strong compared to the best gradient-based learners. (iv) With regard to the debate [30, 4] about whether adaptation is necessary for meta-learning in vision, the comparatively unreliable performance of ANIL and the feed-forward learners suggests that feature adaptation is indeed important to achieve high performance in adaptive channel coding.

### 4.4 Impact of Number of Tasks on Meta-Learning Performance

Few studies have investigated how meta-learner performance depends on the number of meta-training tasks. This is partly because it is not straightforward in standard vision benchmarks to generate enough tasks (objects to recognize) to saturate performance with respect to task number. For our coding benchmark, we can sample an unlimited number of tasks (unique channels) to investigate this. We consider both within- and across-family scenarios where models are trained on AWGN with expanded range (SNR $\in$ [-5, 5]), and tested on all 4 family types. For each learner we evaluate $n \in \{100, 50, 20\}$ unique tasks (domains/channel parameters) while keeping the total number of messages (categories to recognise), and total number of unique samples fixed.

Fig. 7 shows performance as a function of the number of unique training channels, averaged over all four families of testing channels. Overall, most of the learners except for Reptile, which has relatively high BER in all settings, experience degradation in accuracy as the number of unique domains decreases. The ranking of absolute BER values remain constant as the number of domains changes, While the normalized BER curves suggest some meta-learners, e.g. BOIL, MetaSGD, and MetaCurvature, experience more degradation in the sparse task regime than others e.g. CAVIA and MAML FO.

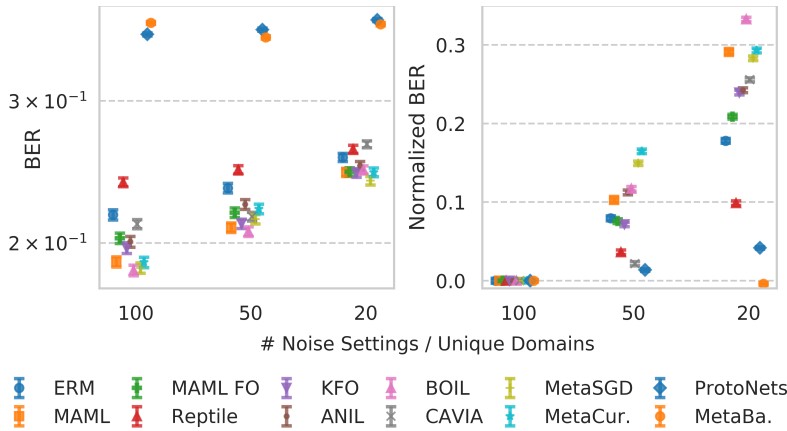

**Figure 7:** Dependence of error rate on number of unique domains (channel conditions). Left: Absolute values. Right: Normalized by BER of the corresponding learner when there are 100 domains.

## 5 Discussion

**Summary** We presented a new meta-learning benchmark based on channel coding, a real-world and practically important problem that lends itself to meta-learning. We summarize by answering the questions we posed in the introduction. **Q1:** *How vulnerable are existing meta-learners to underfitting when trained on complex task distributions?* Building on our task-distribution breadth metric, we quantified this relationship in (Fig. 3, 4). Our results show a clear degradation in performance with breadth, mirroring results in robotics [40]. However, compared to zero-shot transfer of vanilla ERM, the benefit provided by meta-learning can increase with distribution complexity. **Q2:** *How robust are existing meta-learners to task-distribution shift between meta-train and meta-test task distributions?* While absolute performance does decrease under distribution shift (Fig. 6), by comparing our task distribution shift metric with relative improvement over ERM in Fig. 5 (Mid., Right), we showed that performance *margin* of adaptive neural decoders actually tends to improve with distribution-shift. **Q3:** *How much can meta-learning benefit in terms of transmission error-rate on a real radio channel?* Our results show that a few pilot codes are sufficient for a meta-learned adaptive decoder to provide a substantial 58% and 30% reduction in error rate compared to the standard neural decoder and classic Viterbi decoder respectively in real-world channel. This confirms the practical value of this benchmark, as advances can translate to substantial improvements in comms performance.

**Benchmark** Our MetaCC benchmark provides a number of benefits to the community going forward: (i) It provides a systematic framework to evaluate future meta-learner performance with regards to under-fitting complex task distributions [40, 39] and robustness to train-test task distribution shift [12, 37, 40] that is ubiquitous in real use cases such as sim-to-real. Both of these are crucial challenges which must be addressed for meta-learners to be of practical value in real applications. (ii) MetaCC has the further advantage of being independently *elastic* in every dimension. Future studies can thus use it to study impact of number of tasks, instances or categories; dimension of inputs; difficulty of tasks, width of task distributions and train-test distribution shift. Unlike existing saturated small toy benchmarks [20], or large unwieldy benchmarks [37], these properties make it suitable for the full spectrum of research from fast prototyping to investigating the peak scalability of meta-learners. (iii) By addressing rapid adaptation to new *domains*, MetaCC complements existing multi-task focused meta-learning benchmarks. This means that meta-learners are challenged to beat strong baselines including ERM and classic Viterbi algorithms. With regard to robustness, this will allow meta-learners to ultimately be compared directly against methods that improve the ERM baseline through improving robustness to domain-shift [11]. (iv) Finally, MetaCC directly instantiates a task of significant real-world importance, where advances will immediately impact future communications systems [6, 1].

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
