# OpenReview forum: "A Channel Coding Benchmark for Meta-Learning"
_NeurIPS.cc/2021/Track/Datasets_and_Benchmarks/Round2 — NeurIPS 2021 Datasets and Benchmarks Track (Round 2)_

### Official Review · Reviewer_fjKK · 2021-09-20
**Novel meta learning setting with controllable task distribution**

**Rating:** 5
**Confidence:** 2
**Clarity:** The writing is pretty clear.

**Strengths:**

The ability to control the task distributions seems to be the key contribution of this work. The authors argue that current meta learning methods perform poorly in real-world settings in which often involve very broad multi-modal distributions and the ability to overcome distribution shift. Furthermore, current meta learning benchmarks do not admit control over the distribution, so this seems like a novel contribution, and indeed a useful contribution so long as this is a good proxy for the aforementioned distribution issues.

**Weaknesses:**

While this seems like a step in the right direction in terms of solving the problem that the authors use to motivate this benchmark (i.e., distribution breadth, multimodality, and shift), I’m concerned about how good of a proxy this benchmark really is for other real-world modalities which might be complex, high dimensional, and highly multimodal distributions – which is, as I understand it, the goal of the work. Figure 4 seems to suggest that as breadth increases, many methods either don’t see a change in performance (measured by improvement over ERM) or actually improve with increased breath, which seems to suggest that even in this modality, the purported distribution breadth problem isn’t a real issue compared to how ERM fares, which muddles the picture. On the other hand, Figure 3 does show a decrease in absolute performance (measured by BER) when moving to the multimodal setting, however the plot is in log scale and the differences seem small to me (especially for the memory task subfamily), which, if I’m interpreting the scale of the differences correctly, suggests that the distribution breadth problem is real in this context, to some degree, but the impact is marginal. I would be somewhat more convinced if the authors provided a multimodal setting in which performance was dramatically worse.

**Additional Feedback:**

Figure 7 was confusing to me since the x-axes seem to be reversed.

**Correctness:**

The claims seem correct, and the experiments seem to be performed in a reasonable way.

**Documentation:**

Sufficient detail and code is provided with sufficient instructions.

**Ethics:**

No ethical issues to note.

**Relation To Prior Work:**

The authors discuss how prior meta learning benchmarks do not admit control over the task distribution and that they are weak proxies for real-world task distributions.

**Summary And Contributions:**

This work provides a benchmark for meta learning comprising the channel coding problem, which, unlike previous benchmarks, enable the study of distribution shift between train/test sets and the breadth of the distribution over tasks since the channel coding problem admits direct control over these properties (this isn’t possible for many other real-world modalities, e.g., vision). The authors evaluate several meta learning methods on their benchmark.

---

> ### Author Response · Authors · 2021-09-29
> **Response to Reviewer fjKK**
>
> Thank you for your comments.
>
> **Is the benchmark a proxy for real high-dimensional, complex and multi-modal problems?** We would like to emphasize the fact that channel coding is a real-world high-importance problem, which is complex, high dimensional, and is subject to multimodal distributions. Complexity of decoding depends on the coding scheme used and the channel noise characteristics. Dimensionality depends on the input message length. Uniquely these properties allow exploring a range of dimensionalities and complexities within a single benchmark.
>
> **Dependence on training distribution breadth (Fig 3&4)?** Please note that our results in Fig 3&4 are aggregates over many sampled testing scenarios with randomly drawn continuous channel family parameters (e.g., SNR, multipath delay), which naturally reduces some gaps.
> As requested by this reviewer (“*I would be somewhat more convinced if the authors provided a multimodal setting in which performance was dramatically worse*.”) We plot the same results again but without averaging as shown in Fig 10 in Appendix. As can be seen from the results, BER gap between unimodal settings and mixed (multimodal) settings can be as high as 0.06, 0.02, and 0.04 in Bursty, Memory, and Multipath channels respectively. This represents a significant drop in accuracy for channel coding, which in real-life can translate to considerable drop in quality of service experienced by mobile phone users. Note that the difficulty of the problem is non-uniform across x-axis. For example, in the case of bursty channel, the lower the SNR-B is, the harder the problem becomes. While in the case of Memory channel, the alpha value determines how insignificant the memory in noise is.
> We also reiterate that while the dependence on training distribution breadth may be moderate, the dependence on train-test distribution shift (Fig 5 middle, Fig 6) can be severe. The ability to control and quantify the impact of these two facets of a meta-learning problem is a major contribution of our benchmark, without which drawing such conclusions is not straightforward.

---

### Official Review · Reviewer_sqKP · 2021-09-20
**A Nice Dataset on Meta Learning in the Channel Coding Task**

**Rating:** 6
**Confidence:** 2
**Clarity:** The paper is clearly written with det…

**Strengths:**

1) The authors provide practical measurements to quantify the shift of task distributions,
including train-test shift and a diversity score, which are designed specifically for
channel coding task. This is helpful to evaluate the variability of meta-learning models.
2) The dataset covers 5 families of channel models. Besides 4 synthetic ones (Additive White
Gaussian Noise, Bursty, Memory noise, Multipath interference channel), there is a family of
real-world data from radio test-bed. The collection of these data is quite valuable to researchers.
3) A wide range of meta-learning algorithms are concretely evaluated with the dataset,
and the impact of the distribution is carefully studied and discussed by the authors.

**Weaknesses:**

The dataset seems to be limited to the domain of channel coding, but it shouldn't be a big problem as a paper.


**Additional Feedback:**

N/A

**Correctness:**

The dataset constructions and evaluation looks correct and extensive to the reviewer.

**Documentation:**

The basic documents are good to use.

**Ethics:**

The review is not aware of any ethical concerns.

**Relation To Prior Work:**

To the best of the reviewer's knowledge, the literature review is sufficient and looks clear.

**Summary And Contributions:**

This paper introduces a meta learning dataset on a specific task: channel coding.
Complementing to existing ones, the dataset helps the model developers to measure
1) The chance of underfitting;
2) The robustness of the model, in terms of distribution shift.

---

> ### Author Response · Authors · 2021-09-29
> **Response to Reviewer sqKP**
>
> We thank this reviewer for the supportive comments.
>
> **Limited to Channel Coding** Given there is a minor concern raised, even though “*it shouldn't be a big problem as a paper*”, we would like to briefly reiterate the contribution and scope of this paper (which we expect the reviewer may already be well aware of). As has been pointed out by reviewer yh1a, channel coding is not only an important practical application in modern communication systems, but also provides an unique opportunity for research of important topics in meta-learning, e.g. impact of train-test distribution shift and complexity of training distribution on meta-learner’s performance. Furthermore, our channel coding benchmark is suitable for the full spectrum of research from fast prototyping to investigating the peak scalability of meta-learners. Being independently elastic in every dimension, it can thus be used to study impact of number of tasks, instances or categories; dimension of inputs; difficulty of tasks, width of task distributions and train-test distribution shift. To conclude, the potential impact of this benchmark and dataset is not limited to the domain of channel coding.
> Furthermore, to the extent that underpinning research in meta-learning algorithms aims to provide general purpose sample-efficient learners (rather than vision specific ones), it is important for the community to have a diverse suite of benchmarks to measure research progress. Thus differences in conclusions between our benchmark and vision benchmarks are important data points for the community. EG: Metric-based learners (ProtoNet, MetaBaseline and SUR-Proto) perform poorly here, suggesting that metric-learners may not be good general purpose few-shot learners, despite being competitive in vision.

---

### Official Review · Reviewer_yh1a · 2021-09-21
**Channel Coding Benchmark**

**Rating:** 7
**Confidence:** 4
**Clarity:** The paper is very clearly written.

**Strengths:**

-The paper uses both real and synthetic data and studies sim-to-real transfer, which is of practical importance to the communications engineering community.
-The paper demonstrates somewhat surprising results, in contrast to recent findings with the image-based meta-learning community.
-Because of the benchmark's simplicity, rigorous, exhaustive experiments can be performed.
-The benchmark offers clear means by which to control the difficulty of the tasks, and suggests metrics to measure similarity between them.


**Weaknesses:**

-This is not the only benchmark that allows precise control over the generative distribution. For example: synthetic image benchmarks offer similar capabilities but stay in the image domain. Admittedly, I am not aware of any that convincingly demonstrate sim2real transfer in the image domain.
-The benchmark does not evaluate the performance in the case where new classes are introduced at test-time, therefore it is not as broad as some other benchmarks (e.g. MetaDataset). Similarly, the benchmark uses fixed ways and shots.
-I do not see why the authors focused only on "meta-learning" based solutions when this is clearly a few-shot learning problem. I would like to see comparisons to other non-meta-learned algorithms (for example https://arxiv.org/pdf/2003.09338.pdf does not use episodic training).



**Additional Feedback:**

None.

**Correctness:**

-It seems generally correct to me. However it is not clear exactly what is being learned by the decoder. Is the decoder meant to learnt he structure of the channel (e.g. a decoding scheme that works well for the channel models and all possible codes - c(b) or does the decoding scheme also (meta) learn the code structure as well?). In short: does the decoding scheme generalize to all codes or is there a danager of overfitting to particular code structures?

**Documentation:**

Git repo seems reasonable.

**Ethics:**

None.

**Relation To Prior Work:**

This is the first such benchmark applied to channel coding as far as I know.

**Summary And Contributions:**

The paper proposes to measure the performance of various meta-learning algorithms applied to a new channel coding benchmark.
Not only is channel coding an important practical application in modern communication systems, but various transmission channel models allow for precise control of the difficulty of the tasks. This is in contrast to image-classification benchmarks where the generative process is not well controlled and poorly understood. By combining several synthetic channel models with real-world data the benchmark also investigates a "sim-to-real" performance of various approach and compares them to strong baselines from channel-coding/information theory. In contrast with recent findings in the image-based meta-learning community, convincing evidence of the benefit of meta-learning is demonstrated, and impressive performance is realized.

---

> ### Author Response · Authors · 2021-09-29
> **Response to Reviewer yh1a**
>
> We appreciate this reviewer’s positive comments.  We would like to respond to a few details pointed out by the reviewer below.
>
> **Re: Synthetic image benchmarks.** We agree with the reviewer that there exists synthetic image benchmarks that allow control of the generative process (such as Causal3DIdent, https://arxiv.org/abs/2106.04619 and CLEVR). However, we would like to emphasize that our benchmark offers the unique opportunity to *quantitatively measure* the resulting breadth of a task distribution, and train-test distribution shift. This is not straightforward in image benchmarks.
>
> **Re: New classes at test time? Fixed shot/way?** (1) Yes, our current evaluation focuses on novel domain, rather than novel category evaluation because this corresponds to the realistic use case for channel coding, with a closed set of categories implicitly defined by the choice of message length. However, if desired by future research, the framework itself trivially supports evaluation of novel classes at meta-test time simply by holding out some during meta-training. (2) We do use fixed-shot for simplicity (because it corresponds to the realistic use case of channel coding, where there will be a fixed proportion of pilot codes that correspond to the support set, but again this is easy to relax if desired). However we emphasize that it is not fixed-way in the sense of typical few-way vision benchmarks. It is always full-way in that recognition is always performed in the full label-space. EG: If message length is 10 bits, it always performs 2^10 = 1024-way recognition.
>
> **Re: Non-meta-learned alternatives?** Thanks. We note that our ERM baseline is very similar to standard non-meta-learned few-shot methods such as the famous Baseline++ in CloserLook (https://arxiv.org/abs/1904.04232). We thank the reviewer for also suggesting SUR as a strong non-meta few-shot competitor. Please note that SUR is designed for the multi-source cross-domain few-shot setting. It inputs a small set of features trained from each source domain, and learns a fused feature on the target domain support set, before applying ProtoNet. This means that it is relevant to our mixed-family training condition (each channel family corresponds to one SUR input feature). It is not directly relevant to our other experiments as our domains are continuously paramaterized within-family, and so can not provide a discrete set of features for fusion. The original SUR (https://arxiv.org/pdf/2003.09338.pdf) builds on the metric-learner ProtoNet, which we observed performed poorly in our initial experiments. We therefore also propose a novel variant of SUR that builds on ERM rather than metric-learning. Specifically, an ERM model is trained on each source channel family, and the SUR feature selection strategy is applied on the support set in the target domain to fuse these ERM models. We denote these variants as SUR-ProtoNet and SUR-ERM respectively. The results are summarized below, and in Fig 3 of the revised paper. It can be seen that the original SUR Protonet performs poorly as per ProtoNet itself. However, our new SUR-ERM performs well on the benchmark. Thanks for this good suggestion, and inspiring a new algorithm.
>
> | Channel Type | Bursty | Memory | Multipath |
> |--------------|--------|--------|-----------|
> |     ERM      |  0.15  |  0.07  |    0.16   |
> |     MAML     |  0.15  |  0.06  |    0.16   |
> |   SUR ERM    |  0.11  |  0.06  |    0.16   |
> |  SUR Proto.  |  0.42  |  0.38  |    0.41   |
>
> **What is learned by the decoder?** A neural decoder is designed to learn from training data to decode noisy messages without explicitly knowing either the (i) channel model, or (ii) the encoding scheme. The decoder needs to learn the scheme used by the unknown encoder, and it should directly generalize across realisations of the channel noise model, and across all 2^MessageLength possible messages. In this sense we follow prior work [14,15]. However prior work has considered the channel to be unknown but fixed.
>
> In this paper, we relax the assumption of a fixed channel type, and make the decoder adaptive to new channels (corresponding to the case of a moving transmitter experiencing different types and strengths of interference). By meta-learning the decoder, it should gain the ability to adapt to a new channel condition using a small support set (AKA: pilot codes in comms). In short: The decoder should generalize directly to all messages, and adapt rapidly to new channels. We do still consider the encoder itself to be fixed, as this is set by slow moving communications standards bodies such as ITU in practice, and followed by comms industry as a common practice. (Practically, communications systems can employ a set of neural decoders, each corresponding to an encoder.) However, the framework could also easily be extended to study meta-learning of cross-encoder adaptation if this is of academic interest.

---

### Comment · Area_Chair_gzwL · 2021-09-01
**Broken URL**

Dear authors,

the provided URL to the repository https://github.com/ruihuili/MetaCC is invalid.

Please provide a working URL or clarify that all data/code/information necessary for reproduction is available in the supplementary material.

---

> ### Author Response · Authors · 2021-09-01
> **Fixed URL**
>
> Dear Area Chair,
>
> Sorry for the inconvenience, we had to relocate the repo to https://github.com/MetaCoCo/MetaCC.git, but recently  https://github.com/ruihuili/MetaCC.git is also back online. Both links contain the same code. The folder MetaCC-source_code.zip in supplementary material contains all code needed for reproduction. Thank you

---

> > ### Comment · Area_Chair_gzwL · 2021-09-01
> > **Fixed URL**
> >
> > URL works now, thank you for the quick fix.

---

### Decision · Program_Chairs · 2021-10-09

**Decision:**

Accept

**Comment:**

Dear authors,

Thank you for submitting your paper and addressing the issues raised by the reviewers.

The paper proposes a new benchmark for meta-learning based on channel coding. The reviewers commend the practical relevance of the considered task as well as the possibility to directly control the difficulty of the task. Limitations include the lack of newly emerging classes at test time and the fixed shot/way setting of the benchmark.

The scores of the reviews are 7,6,5. The main criticism of reviewer 3, was the relation to real world problems, which was adequately addressed by the authors in their rebuttal. Considering this as well as the high confidence of reviewer 1 giving a score of 7 is sufficient to accept the paper.